# Scaffold Pore Curvature Influences ΜSC Fate through Differential Cellular Organization and YAP/TAZ Activity

**DOI:** 10.3390/ijms23094499

**Published:** 2022-04-19

**Authors:** W. Benton Swanson, Maiko Omi, Seth M. Woodbury, Lindsey M. Douglas, Miranda Eberle, Peter X. Ma, Nan E. Hatch, Yuji Mishina

**Affiliations:** 1Department of Biologic and Materials Science, Division of Prosthodontics, University of Michigan School of Dentistry, Ann Arbor, MI 48109, USA; wbentons@umich.edu (W.B.S.); maikoo@umich.edu (M.O.); woodbuse@umich.edu (S.M.W.); lmdougla@umich.edu (L.M.D.); eberlemi@umich.edu (M.E.); mapx@umich.edu (P.X.M.); 2Macromolecular Science and Engineering Center, College of Engineering, University of Michigan, Ann Arbor, MI 48109, USA; 3Department of Materials Science and Engineering, College of Engineering, University of Michigan, Ann Arbor, MI 48109, USA; 4Department of Biomedical Engineering, College of Engineering, University of Michigan, Ann Arbor, MI 48109, USA; 5Department of Orthodontics and Pediatric Dentistry, University of Michigan School of Dentistry, Ann Arbor, MI 48109, USA; nhatch@umich.edu

**Keywords:** tissue engineering, scaffolds, regenerative medicine, mechanotransduction, mesenchymal stem cell

## Abstract

Tissue engineering aims to repair, restore, and/or replace tissues in the human body as an alternative to grafts and prostheses. Biomaterial scaffolds can be utilized to provide a three-dimensional microenvironment to facilitate tissue regeneration. Previously, we reported that scaffold pore size influences vascularization and extracellular matrix composition both in vivo and in vitro, to ultimately influence tissue phenotype for regenerating cranial suture and bone tissues, which have markedly different tissue properties despite similar multipotent stem cell populations. To rationally design biomaterials for specific cell and tissue fate specification, it is critical to understand the molecular processes governed by cell-biomaterial interactions, which guide cell fate specification. Building on our previous work, in this report we investigated the hypothesis that scaffold pore curvature, the direct consequence of pore size, modulates the differentiation trajectory of mesenchymal stem cells (MSCs) through alterations in the cytoskeleton. First, we demonstrated that sufficiently small pores facilitate cell clustering in subcutaneous explants cultured in vivo, which we previously reported to demonstrate stem tissue phenotype both in vivo and in vitro. Based on this observation, we cultured cell-scaffold constructs in vitro to assess early time point interactions between cells and the matrix as a function of pore size. We demonstrate that principle curvature directly influences nuclear aspect and cell aggregation in vitro. Scaffold pores with a sufficiently low degree of principle curvature enables cell differentiation; pharmacologic inhibition of actin cytoskeleton polymerization in these scaffolds decreased differentiation, indicating a critical role of the cytoskeleton in transducing cues from the scaffold pore microenvironment to the cell nucleus. We fabricated a macropore model, which allows for three-dimensional confocal imaging and demonstrates that a higher principle curvature facilitates cell aggregation and the formation of a potentially protective niche within scaffold macropores which prevents MSC differentiation and retains their stemness. Sufficiently high principle curvature upregulates yes-associated protein (YAP) phosphorylation while decreased principle curvature downregulates YAP phosphorylation and increases YAP nuclear translocation with subsequent transcriptional activation towards an osteogenic differentiation fate. Finally, we demonstrate that the inhibition of the YAP/TAZ pathway causes a defect in differentiation, while YAP/TAZ activation causes premature differentiation in a curvature-dependent way when modulated by verteporfin (VP) and 1-oleyl-lysophosphatidic acid (LPA), respectively, confirming the critical role of biomaterials-mediated YAP/TAZ signaling in cell differentiation and fate specification. Our data support that the principle curvature of scaffold macropores is a critical design criterion which guides the differentiation trajectory of mesenchymal stem cells’ scaffolds. Biomaterial-mediated regulation of YAP/TAZ may significantly contribute to influencing the regenerative outcomes of biomaterials-based tissue engineering strategies through their specific pore design.

## 1. Introduction

Tissue engineering aims to repair, replace, and restore biologically functional tissues in the human body through synergistic applications of biomaterials, cell signaling moieties, and cell sources capable of regeneration [1,2]. Biomaterials serve as an artificial, temporary extracellular matrix (ECM) to organize regeneration [3]. In recent years, specific criteria of biomaterials constructs have been identified which contribute to their success as tissue engineering matrices [4]. Nanofibrous and textured biomaterial implants result in increased protein adsorption and cell adhesion [5], proliferation [6,7,8,9], and differentiation [10,11,12]. Highly porous implants facilitate mass transfer and host integration [13,14,15,16]. Improved fabrication protocols have facilitated advancements in highly tunable constructs of various stiffness [17,18], controlled drug delivery [19,20] and stimuli responsive behavior [21,22]. Such findings enable us to elucidate the contribution of each of these factors towards tissue fate. The grand challenge identified in currently available biomaterials, hindering their widespread clinical use, is improving the predictability of regenerative outcomes [23]. To accomplish this, an intimate understanding of the cell-biomaterial interface is critical for purpose-designed regenerative biomaterials [24].

Mesenchymal stem cells (MSCs), capable of the maintenance, repair and regeneration of skeletal tissues due to their multipotency and proliferative capacity, represent an ideal cell source for skeletal regeneration [25,26,27,28]. Our group is particularly interested in two MSC sources: the cranial suture (suture mesenchymal stem cells, SMSC) and long bone (bone marrow mesenchymal stem cells, BMSC)—each representing a unique regenerative potential. Craniosynostosis is a debilitating developmental disease of the cranial suture mesenchyme, characterized by premature depletion of the SMSC population, resulting in stunted craniofacial growth and increased intracranial pressure that has the potential to impact both skull and brain growth [29,30]. In a tissue engineering approach to treating craniosynostosis, the ideal biomaterial construct would maintain the stemness of implanted MSCs, acting as a stem cell reservoir to normalize craniofacial growth and development throughout childhood and adolescence [28,31,32]. In contrast, in the case of alveolar bone regeneration–for example guided tissue regeneration in preparation for dental implant placement or osseous surgery for periodontal regeneration—the desirable regenerative outcome is for MSCs to predictably differentiate to form robust bone [33,34,35].

Our group previously demonstrated unique scaffold microenvironments adequate for osteogenic differentiation and suture stem cell maintenance, modulated by scaffold macropore size, as summarized in Figure 1 [27,28]. Optimized nanofibrous, macroporous scaffolds of discrete, uniform pores less than 125 μm in diameter recapitulate aspects of the cranial suture mesenchyme and maintain stemness of SMSCs. On the other hand, nanofibrous macroporous scaffolds of sufficiently large pore diameter, greater than 250 μm, facilitate robust osteogenic differentiation and vascularization of a mature extracellular matrix. We established that porosity is a key consideration in scaffold design using a well-controlled scaffold fabrication platform; however, the underlying mechanisms by which cells respond to their macroporous microenvironment have yet to be elucidated.

This schematic illustrates that the same cell type is capable of multiple regenerative fates, guided by the biomaterial microenvironment, which acts as an exogenous factor to guide regeneration. Building from these phenotypic observations in vivo and in vitro, we aimed to determine the molecular mechanism by which scaffold pores modulate cell fate. In addition to mechanistic insight at the cell-biomaterial interface provided in the present study, we also assess potential differences between cranial neural crest-derived (SMSC) and trunk-derived (BMSC) MSCs of similar function but distinct embryonic origin.

Skeletal organs respond to mechanical stimuli [36,37]. For example: a tennis player’s dominant arm demonstrates increased cortical thickness and bone mineral density due to increased repeated loading [38]. Strength training is demonstrated to slow or reverse the progression of osteoporosis [39], and the cyclic tensile strain of the cranial suture causes suture maturation and differentiation [40,41]. Traditionally, osteocytes have been thought to be the mechanosensitive organ of bone [42]. More recently, MSCs have also been demonstrated to respond to their mechanical microenvironment, and correlation has been drawn between mechanical stress and cell differentiation fate [43,44]. Cells‘ mechanical environments include stiffness [45], cell adhesion area [46,47], compression [48] and tension [49,50]. Each of these factors, among others, may be designed in a biomaterial scaffold to guide MSC fate. The cytoskeletal machinery of MSCs, including integrins, focal adhesion kinase, paxillin, talin, vinculin and others, transduce cytoskeletal strain to the nucleus [51]. Critically, the organization of the cytoskeleton mediates the transfer of microenvironmental strain through the cell and guides its response to loading.

Building on our previous work, we sought to determine the role of scaffold macropore size to modulate differentiation and stemness of skeletal and craniofacial skeletal MSCs, given their mechanosensitivity. Specifically, we aim to address the degree to which principle curvature of macropores regulates differential cell fate and provide evidence to establish our major hypothesis that YAP/TAZ signaling is an important regulator of MSC fate in biomaterial tissue engineering scaffolds. The overall goals were to: (1) elucidate the cause-effect relationship of principle curvature on cellular and nuclear organization; (2) elucidate the significance of cytoskeletal strain on cell fate using a modulator of actin polymerization; (3) identify molecular pathways differentially upregulated soon after scaffold seeding that may transduce curvature-mediated cell fate; and (4) validate these findings with a pharmacologic modulation of the target pathway. Overall, here we report that steep principle curvature in sufficiently small diameter pores (<125 μm diameter) inhibits YAP/TAZ activation and favors cytosolic YAP phosphorylation (inactivation), contributing to maintenance of stemness. On the other hand, sufficiently shallow principle curvature facilitates YAP/TAZ activation and nuclear translocation as early as 48 h after seeding and is in-part responsible for curvature-mediated osteogenic differentiation. Together, our work shows that we can take advantage of the mechanosensitivity of MSCs to engineer tissue fate using micro-scale features of a biomaterial by using advanced scaffolding technology–to drive towards stemness in the case of craniosynostosis treatment, or towards differentiation in the case of bone augmentation.

## 2. Results

### 2.1. Discretely Controlled Macropore Diameters Modulate Principle Curvature in Nanofibrous Scaffolds

Previously, we reported that pore size modulates tissue phenotype in vitro and in vivo [27,28]. Sufficiently large pores (>250 μm diameter), with sufficiently low (shallow) principle curvature, facilitated an osteogenic fate towards differentiated bone. Sufficiently small pores (<125 μm diameter), with sufficiently steep principle curvature, facilitates the maintenance of stemness and prevented osteogenic differentiation. In vivo, we demonstrated that pores >250 μm diameter facilitated robust vascularization in terms of vasculature in-growth, vasculature density, vasculature maturity, and more mature collagen in the extracellular matrix [27,28]. Based on these findings, we hypothesized that biomaterial curvature specifically may modulate cell fate. We also sought to investigate in more detail the parameters of our scaffolds.

Scanning electron micrograph (SEM) images demonstrate the uniform spherical macropores of our scaffolding platform, fabricated from poly (L-lactic acid) (PLLA, Figure 1A), consistent with our previous work [20,27,28]. The size range of macropores is controlled by a sugar sphere porogen method, summarized schematically in Figure 2, where molecular sieves select for porogens of specific diameter ranges: large (250–425 μm) and small (60–125 μm). Interconnections between macropores allow for nutrient and waste exchange, as well as cell migration and vascular ingrowth. The relative pore surface area that is missing due to interconnections between adjacent pores is independent of pore size [28]. In addition, the compressive modulus does not vary significantly by pore size [28] and is sufficiently stiff so that differential cellular response is attributed to morphology, rather than stiffness of the constructs [52].

We determined the average principle Gaussian curvature to be 0.0157 μm^−1^ and 0.0058 μm^−1^ for large and small macropores, respectively, based on image analysis (Figure 1B). The solid angle subtended by a cell (a measure of the area of the segment of a unit sphere, centered at the apex, that an object covers) approximates the curvature experienced by a single cell as a result of pore diameter. The solid angle is defined as:$$Solid\;Angle\;\left(K\right)=\frac{1\;sr}{r^{2}}.$$

We approximated the surface area of an MSC as an elipsoid with 10 μm height and 20 μm width, resulting in surface area of 536 μm^2^. Based on this, the average solid angle subtended by an MSC is 0.0189 sr and 0.248 sr, for large and small macropores, respectively (Figure 1C).

### 2.2. Scaffold Pore Size Facilitates Cell Organization In Vivo and In Vitro

After 4 weeks of subcutaneous implantation in wild type mice, where BMSC and SMSCs had been seeded to scaffolds of uniformly low or high principle curvature (large and small pores, respectively; 5 mm diameter × 1.5 mm height), we assessed the spatial distribution of cells within the scaffold construct for insight as to how pore curvature may differentially modulate tissue organization (Figure 2A). After four weeks we previously observed phenotype differences between engineered tissue constructs based on pore size [28]. Both constructs and cell types demonstrate cells which are well-spread throughout the macropores.

Spatial point pattern analysis (SPPA) computes the estimated statistical spatial density of events relative to an arbitrary field of points within the available space [53,54]. In this case, the events are MSC nuclei as assessed from H&E histology, and available space is the scaffold macropores. We identified an increased tendency of MSCs to cluster near to each other in small pore (S, <125 μm diameter) scaffolds compared to large pore scaffolds (L, >250 μm diameter), consistent between both cell types (Figure 2B) when compared to a Monte Carlo simulation of randomness (96% envelope, AUC_Exp_ − AUC_Theo_ = 0; AUC: Area Under the Curve; Exp = experimental; Theo = theoretical). While both pore size scaffolds facilitate some degree of cell clustering compared to complete spatial randomness, this observation is seen more strongly in pores <125 μm diameter, with a sufficiently steep principle curvature (Figure 2C). Monolayer (flat) two dimensional cell culture is an example of zero principle curvature and complete spatial randomness. Calculation of second-order K-statistic by SPPA also indicates an increased tendency of MSCs to appear clustered compared to a homogenous poison distribution (complete spatial randomness) in the <125 μm diameter pore scaffolds compared to large pore scaffolds (*p* < 0.05), for both cell types, Figure 2D).

Our in vivo findings suggest that scaffold pore size modulates cell and tissue fate, in part, by differential cell organization. In cell-scaffold explants, which we previously demonstrated to facilitate maintenance of stemness and lack of differentiation [28], we observe increased cell clustering (F-statistic) and the tendency of cells to cluster (K-statistic) within scaffold macropores. On the other hand, in cell-scaffold explants, which we previously demonstrated to facilitate robust osteogenic differentiation at this time point [27,28], we observe less cell clustering and an increased tendency towards spatial randomness. Based on this proof-of-principle observation, which correlates tissue phenotype with cellular organization, we next assessed the degree to which scaffold macropore curvature facilitates cell organization soon after seeding in vitro using the same scaffold platform. In vitro assessment allows for high throughput investigation without the influence of the host environment, specifically isolating the modulation of cell fate to variables related to the biomaterial construct, namely pore size.

Consistent with our hypothesis, pores >250 μm diameter (L) consistently facilitate a higher nuclear aspect ratio (NAR) and a more elongated nucleus compared to nuclei in pores <125 μm diameter (S) (Figure 3A,B, L vs. S are significant at each time point and cell type, *p* < 0.001), without significantly affecting nuclear cross-sectional area (Figure 3C). Consistent with our observations in vivo, pores <125 μm diameter facilitate increased clustering within macropores (Figure 3D, L vs. S) and exhibit a decreased nearest neighbor distance (Figure 3E, L vs. S are significant at each time point and cell type). Despite these changes in cell morphology, which are consistent for at least the first week of in vitro culture, the scaffolds are uniformly cellularized and MSCs are well-distributed throughout the volume of the scaffold at 1 weeks’ time (Appendix A).

### 2.3. Cytoskeletal Disruption in Scaffold Macropores Alters Cell Morphology and Gene Expression

Given our observations that macropore principle curvature alters cell distribution and nuclear morphology within scaffold constructs, and strong evidence in the literature to suggest that cytoskeletal morphology affects nuclear activity [43,46,51], we were curious as to the degree to which cytoskeletal integrity modulates curvature-mediated tissue fate. Cell-scaffold constructs were treated with cytochalasin D (CytD), an inhibitor of actin polymerization. CytD caused a rounder cytoskeletal and nuclear morphology, which were more pronounced in constructs with pore diameter >250 μm as compared to pore diameter <125 μm, in a dose-dependent manner (Figure 4A). Gene expression analysis of cell–scaffold constructs treated with 500 nM CytD for 24 h, following 1 week of in vitro culture, highlights that brief cytoskeletal disruption during early cell–scaffold culture affects differentiation (Figure 4B). For both BMSC and SMSCs, large pore constructs treated with 500 nM CytD demonstrate an increase in stemness marker CD44 and trend suggesting a decrease in osteogenic markers RUNX2 and SP7. The treatment of small pore constructs for both cell types demonstrate an unexpected decrease in CD44 expression, indicating that some degree of curvature-mediated cytoskeletal arrangement is likely critical.

### 2.4. Curvature-Mediated Cytoskeletal Strain May Differentially Regulate YAP/TAZ Signaling in Cell–Scaffold Constructs

The actin cytoskeleton and stress fibers resulting from mechanical strain on cells are important mediators between a cell’s mechanical external environment and intracellular signaling processes, specifically via Hippo-YAP signaling (Figure 3). CytD treatment has been described to indirectly inactivate YAP/TAZ signaling and decrease mechano-induced transcription [55,56,57]. Given the gene expression pattern observed in Figure 4 resulting from 24 h of pharmacologic cytoskeletal disruption, we hypothesized the potential involvement of YAP/TAZ signaling in curvature-mediated tissue fate, particularly at early time points following cell seeding. Indeed, we confirmed that CytD treatment induces YAP phosphorylation (inactivation) and cytosolic retention in large pore scaffolds in a dose dependent manner, the combination of which contributes to impaired YAP/TAZ signaling (Figure 5).

We next assessed gene expression related to mechanotransduction to further probe our hypothesis that YAP/TAZ may be a driver of curvature-induced cell fate. Three macroporous scaffold constructs were fabricated: small (S, 60–125 μm diameter), medium (M, 125–250 μm diameter) and large (L, 250–425 μm). Unrestricted hierarchical clustering by Pearson correlation identified constructs with pore size >250 μm as having a distinct differentiation trajectory from other cell–scaffold constructs beginning at 24 h post-seeding, most distinct by 48 h (Figure 6A). We observe curvature-mediated induction of genes directly targeted by and involved in YAP/TAZ signaling (CTGF, YAP, RHOA), and genes involved in focal adhesion kinase signaling (VCL, ZYX, TLN1, TLN2, RHOA) and nuclear proteins (LMNA). Most notably, we observe the upregulation of CTGF expression, a direct target of YAP, as well as increased YAP1 expression in scaffolds with sufficiently low principle curvature (Figure 6B). Western blot analysis of cell–scaffold constructs corroborated with CTGF gene expression, demonstrating increased YAP/TAZ phosphorylation in constructs with pore size <125 μm diameter (S) by day 3 of in vitro culture, compared to >250 μm diameter pore constructs (L) and monolayer (F) control (Figure 6C,D). Monolayer control (F) is considered as infinitesimally low principle curvature. Pore diameters greater than 250 μm showed a distinct trajectory of YAP/TAZ activation, while the difference between 125–250 μm and 60–125 μm is less dramatic, but still distinct with time, facilitating YAP/TAZ inactivation.

### 2.5. Reproducing Curvature-Mediated Cell Organization with a Macropore-Mimetic Hemisphere Platform to Assess Differential YAP Regulation

We sought to recapitulate the three-dimensional macropore microenvironment with a well-controlled platform that lent itself to confocal laser microscopy imaging. We fabricated a macropore-mimetic hemisphere (MMH) from polydimethylsiloxane (PDMS), an optically clear silicon-based polymer, by soft lithography techniques (Appendix A). Master molds were designed with pore hemispheres to match the average pore diameters observed in our scaffolds by SEM (Appendix A–D). PDMS was cast (Appendix A), and constructs were seeded with MSCs for up to one week of in vitro culture (Appendix A). No cytotoxicity was observed. To ensure that the PDMS MMH platform accurately reproduced biologic phenomena we observe in scaffold constructs, the stiffness of PDMS was matched to our PLLA scaffolds by optimizing its synthesis (Appendix A). Confocal laser microscopy and SEM imaging of PDMS constructs cultured with MSCs for 1 week in vitro serve as a proof of concept that these hemispheres regulate cell organization and recapitulate macropore curvature-induced organization (Figure 7).

We assessed the regulation of YAP/TAZ signaling in BMSCs cultured in MMH constructs for up to 1 week in vitro by immunofluorescence staining of YAP phosphorylation (Figure 8). At 24 h, no significant differences in YAP phosphorylation are observed. By 48 h, pYAP expression is increased in small MMH pores and nearly depleted in large pore constructs. This differential YAP/TAZ regulation is maintained at 1 week and is consistent for both BMSC and SMSC (Appendix A), corroborating the Western blot evidence shown in Figure 6.

CD146 expression, a skeletal progenitor cell marker, follows a similar pattern to pYAP at 48 h, correlating stemness to YAP activity (Figure 9). Sufficiently steep principle curvature facilitates cell clustering, as observed in vivo and in vitro, and is recapitulated in the MMH model. When observed by cross-section from three-dimensional confocal laser microscopy, the most intense CD146 expression originates from the epicenter of the cell cluster and dissipates towards the periphery, suggesting a potential progenitor/stem cell-protective niche facilitated by curvature-induced clustering in sufficiently small diameter macropores. The same is observed for pYAP expression at 48 h and 1 week in vitro (data not shown).

### 2.6. Curvature Differentially Regulates YAP Nuclear Translocation and Its Cytosolic Phosphorylation In Vitro in Cell-Scaffold Constructs

Given our promising results suggesting curvature-mediated YAP inactivation in constructs with sufficiently steep principle curvature in the MMH model, we next sought to quantitatively assess YAP nuclear accumulation (indicates YAP/TAZ activation) and YAP phosphorylation (indicates YAP/TAZ inactivation) directly in PLLA macroporous scaffolds. Constructs were cultured in vitro for up to three weeks and assessed for total YAP protein (tYAP, Figure 10A) and YAP phosphorylation (pYAP, Figure 10B) in histologic sections. Based on nuclear colocalization analysis of tYAP, we demonstrate that pores >250 μm facilitate YAP nuclear localization by 48 h, which is maintained at peak levels for up to 1 week (Figure 10C). Nuclear localization of YAP remains higher in sufficiently low principle curvature constructs compared to pores <125 μm in diameter where YAP nuclear translocation remains at basal levels, for at least three weeks. Consistent with observations from the MMH model and Western blot, YAP phosphorylation dramatically decreases in sufficiently low principle curvature constructs at 48 h in vitro (Figure 10D). Low levels of pYAP are maintained for up to three weeks. On the other hand, sufficiently steep principle curvature facilitates the upregulation of pYAP in the first week of culture and is maintained for up to 3 weeks.

### 2.7. Pharmacologic Intervention Highlights the Role of YAP/TAZ Signaling to Regulate Gene Expression Where Curvature-Mediated Cytoskeletal Strain Is the Major Driver of Regenerative Fate

To confirm the role of YAP/TAZ signaling in curvature-mediated engineered tissue fate determination, we tested the ability of a pharmacologic inhibitor and agonist to manipulate curvature-directed tissue fate. Constructs were treated for 5 days with either verteporfin (VP) or lysophosphatidic acid (LPA). VP is a small molecule inhibitor that physically blocks YAP/TAZ from binding its transcriptional co-factors, namely transcriptional enhancer associated domain (TEAD) transcription factors (Figure 3), therefore reducing YAP/TAZ-mediated transcriptional activity (Appendix A). We demonstrate that VP administration abrogates YAP transcriptional activity by showing diminished expression of target gene CTGF and YAP1 in pore size >250 μm. VP simultaneously causes decreased osteogenic RUNX2 expression and the upregulation of stem progenitor cell marker GLI1 (Figure 11A). Interestingly, the expression of CD44, a stem cell marker, was decreased modestly by VP treatment.

The ROCK agonist LPA (lysophosphatidic acid) was used to directly interfere with YAP phosphorylation and increase YAP/TAZ activation in a cytoskeleton-independent manner (Appendix A). LPA is also shown to increase stress fiber density without altering nuclear area [51]. We demonstrate that pharmacologic activation of YAP/TAZ by LPA overcomes inactivation facilitated by a sufficiently steep principle curvature to increase CTGF and YAP transcription in scaffolds with pores <125 μm in diameter. Consistent with our overall hypothesis, LPA administration causes a modest increase in early osteogenic gene expression (RUNX2) and a decrease in stem cell markers CD44 and GLI1 (Figure 11B).

## 3. Discussion

Skeletal tissues, including alveolar bone and the cranial suture, are composed of a rich extracellular matrix (ECM) and are exposed to significant mechanical loads which are transmitted to cells and nuclei [58,59,60]. In their physiologic roles, MSCs of these tissues undergo marked differentiation in response to mechanical stress to form differentiated bone [61], in nearly all physiologic systems [62]. In order to maintain MSC stemness, studies of the cranial suture indicate a mechanoprotective ECM environment, which may function to maintain SMSC stemness and prevent osteogenic differentiation [63]. The means by which biomaterial scaffolds serving as an artificial ECM could modulate this cellular response remains underexplored in the field of tissue engineering. A few recent reports [40,64] describe the roles of mechanotransduction in various biomaterial cell culture platforms such as gelatin methacrylate (GelMA), polyacrylamide (PA) or poly ethyleneglycol (PEG) gels, as well as two-dimensional cell sheets, namely focused on substrate stiffness. Here, we focused on the degree to which geometric macropore curvature in a highly porous scaffold construct regulates YAP/TAZ activity and osteogenic differentiation, in a platform which minimizes stiffness as a confounding variable.

In this study, we have identified the role of YAP/TAZ signaling and modulation of its nuclear translocation in response to fundamental differences in MSC microenvironment in synthetic biomaterial constructs. Macroporous, nanofibrous tissue engineering scaffolds from PLLA have been well-demonstrated in their application to a variety of skeletal tissue engineering applications [3,10,19,20,27,28,65]. Based on our previous data which suggests that pore size mediates engineered tissue fate, we aimed to determine the molecular means by which scaffold principle curvature contributes towards either maintenance of MSC stemness or commitment to a differentiated tissue fate. Our scaffolds are fabricated with well-controlled size of their internal macropores (Figure 1) [65]. Considering the curvature subtended by an individual MSC at the macropore surface, we show that increasing pore size dramatically decreases the solid angle subtended by a cell—a 13-fold difference between our large (250–425 μm diameter macropore) and small pore (60–125 μm diameter macropore) constructs. Using a systematic approach we identified significant alterations in tissue organization, and cell and nuclear morphology resulting from differences in principle curvature (Figure 2 and Figure 3).

Recent evidence suggests MSCs as being highly mechanosensitive [44,50,52]. In addition to force sensation mechanisms, the cytoskeleton is responsible for transmitting extracellular strain to the nuclear envelope [51]. Driscoll et al. demonstrate that cytoskeletal tension regulates nuclear shape and force transmission in MSCs, and cytoskeletal strain transfer is essential for YAP/TAZ pathway activation. YAP/TAZ is a master regulator of mechanotransduction, reliant on the nuclear translocation of YAP for its activation [66]. YAP phosphorylation maintains YAP in the cytosol and is a marker for its proteolytic destruction (Figure 3). Consistent with previous studies, we demonstrate that the inhibition of cytoskeletal machinery by CytD alters cell morphology, abrogates curvature-induced osteogenic differentiation in sufficiently large diameter scaffolds by alleviating cytoskeletal strain, and alters intracellular YAP/TAZ dynamics (Figure 4 and Figure 5) within scaffold macropores.

In our macroporous scaffolds, we provide direct evidence of YAP/TAZ activation at the gene and protein expression levels, which supports our overarching hypothesis that curvature mediates YAP activity and is a driver in determining engineered tissue fate (Figure 6). We designed a three-dimensional macropore mimetic hemisphere (MMH) model allowing for direct visualization of differential YAP inactivation that simultaneously provides mechanistic correlation to the role of cell clustering and its effect on YAP activity in pores with sufficiently steep principle macropore curvature (Figure 7, Figure 8 and Figure 9). These findings and our previous work suggest that SMSC, a craniofacial skeletal progenitor cell population, and BMSC, a long bone skeletal hematopoietic progenitor cell population, respond similarly to curvature by YAP/TAZ modulation. In histologic sections of cell–scaffold constructs we confirm these findings and demonstrate that sufficiently low principle curvature in macropore sizes greater than 250 μm diameter facilitate robust nuclear YAP trafficking and significantly reduce YAP phosphorylation as early as 48 h that is maintained throughout in vitro culture (Figure 10). On the other hand, sufficiently steep principle curvature in macropore sizes less than 125 μm diameter facilitated YAP phosphorylation and cytosolic maintenance to largely inactivate YAP/TAZ signaling (Figure 10). Finally, to confirm these findings, we demonstrate that pharmacologic inhibition of YAP signaling in large pore scaffolds leads to a defect in established curvature-mediated osteogenic differentiation, and pharmacologic YAP activation in small pore scaffolds stimulates premature differentiation and loss of stemness (Figure 11).

The role of the YAP/TAZ signaling axis is established in the context of skeletal progenitor cells and mineralized tissue biology. Kegelman et al. demonstrated that YAP and TAZ deletion in an Osterix-Cre mouse model causes a defect in bone formation resulting in deficient mechanical properties, concluding that YAP/TAZ signaling is a critical mediator of osteoprogenitor function and bone formation [66]. Pharmacologic inhibition with verteporfin in our studies likewise resulted in defective osteogenic function in vivo and in vitro. YAP and TAZ combinatorically promote bone formation in the contexts of development and repair, an area of ongoing investigation [61]. Tang et al. describe Slug/Snail proteins as an intracellular means of stabilizing the YAP/TAZ complex and preventing its proteasomal degradation, enhancing osteogenic differentiation of skeletal progenitor cells and enhancing YAP/TAZ-related gene expression [67]. Mechanical stimulation specifically enhances osteogenic differentiation with no apparent effects on proliferation [68], also promoting YAP dephosphorylation, enabling its nuclear transport and transcriptional activation [69]. To further support similar behaviors between BMSC and SMSCs, Li et al. demonstrate that mechanical tension similarly modulates YAP/TAZ nuclear relocation and loss of stemness concomitant with osteogenic differentiation in SMSCs. The inhibition of YAP signaling suppressed the mechanical tension-induced osteogenesis of SMSCs [70]. In addition to binding to TEAD, YAP/TAZ also binds to RUNX2 (among others) in both BMSC [71] and SMSC [70], a well-established transcription factor involved in early osteogenic differentiation [72]. Finally, TAZ is reported to regulate vitamin D3 receptor expression through the p53/CYP24A1 pathway [73]; vitamin D3 aids in calcium absorption and bone mineralization [74].

The present study focuses on characterizing the response of BMSC and SMSCs to scaffold curvature through mechanistic insight in vitro. An important consideration in tissue regeneration in vivo is the complex milieu of cell types which participate in regeneration. In the context of alveolar bone regeneration, robust vascularization is prerequisite to mineralized bone formation [75]. YAP/TAZ activation in a variety of cancers promotes metastasis and increased angiogenesis; endothelial cell YAP/TAZ activation specifically is attributed to the formation of new blood and lymphatic vessels during development [76].

YAP/TAZ has also been implicated in vasculature sprouting, where deletion of YAP/TAZ leads to a hyper-pruned vascular network in vivo and may be partially responsible for decreased organ and bone mass in Yap/Taz^iΔEC^ mice compared to wild type controls [76]. On the other hand, upregulation of YAP/TAZ may synergistically enhance angiogenesis and vasculature sprouting in engineered tissue constructs, along with inducing osteogenic differentiation. Future investigation in this area is needed to provide additional insight to the response of endothelial cells and others involved with in vivo regeneration in response to the scaffold microenvironment.

Considering the next generation of advanced bioscaffolds, which are rationally designed to guide predictable regeneration, mathematical models that can account for MSC response to various mechanical inputs and their interactions may represent future opportunities for high throughput, advanced scaffolding optimization. As an example, Scott et al. have developed a spatial model of YAP/TAZ signaling which combines multiple mechanical inputs [77]. This type of work, combined with studies like ours which correlate YAP/TAZ activity to transcriptional activity and tissue phenotype, represent an exciting convergence of molecular biology, data science and tissue engineering. It is likely that other mechanotransduction circuits, such as MRTF-SRF [78] and Piezo1/2 [79], may also be at play in the scaffold microenvironment and warrant future investigation. Increasingly, predictable biomaterials-based regenerative outcomes, based on mechanistic studies of the cell–biomaterial interface will facilitate the next generation of tissue engineering matrices.

## 4. Materials and Methods

### 4.1. Materials

Poly (L-lactic acid) (PLLA, Resomer L207S) with an inherent viscosity of 1.6 dL/g was purchased from Boehringer Ingelheim (Ingelheim, Germany); Span80 and sterile-filtered dimethyl sulfoxide (DMSO) were purchased from Sigma (St Louis, MO, USA); tetrahydrofuran (THF) and hexane solvents were purchased from Fisher Scientific (Hampton, NH, USA); D-fructose was purchased from Oakwood Chemical (Estill, SC, USA); mineral oil was purchased from Alfa Aesar (Haverhill, MA, USA). All reagents were used as received unless otherwise noted.

#### 4.1.1. Pharmacologic Reagents

Cytochalasin D (Cayman Chemical, No. 11330, Ann Arbor, MI USA), Verteporfin (Fisher, No. 50-202-9387) and Oleoyl-L-α-lysophosphatidic acid (Fisher, No. J66836LB0, Hampton, NH, USA) were used as received and stored according to the manufacturer’s recommendation.

#### 4.1.2. Antibodies and Immunofluorescence Reagents

The following antibodies were used: YAP (D8H1X) XP Rabbit mAb (Cell Signaling Technologies, No. 14074, Danvers, MA, USA), phosphor-YAP (Ser127, D9W2I) Rabbit mAb (Cell Signaling Technologies, No. 13008, Danvers, MA, USA), CD146/MCAM Rabbit polyclonal antibody (Fisher, 17564-1-AP, Hampton, NH, USA). Additionally, AlexaFluor Phalloidin 488 (Invitrogen, No. A12379, Waltham, MA, USA) was used to visualize the actin cytoskeleton, ProLong Gold Antifade Mountant with DAPI (Invitrogen, No. P36935, Waltham, MA, USA) to visualize nuclei.

#### 4.1.3. Cell Culture Reagents

Bone marrow mesenchymal stem cells (BMSCs) were cultured in DMEM (Invitrogen, 11885-084, Waltham, MA, USA) supplemented with 10% *v/v* fetal bovine serum (FBS, Gibco 10438-026, Waltham, MA, USA) and 1% *v/v* penicillin/streptomycin (P/S, Gibco 15140-122, Waltham, MA, USA). Suture mesenchymal stem cells (SMSCs) were cultured in αMEM (Invitrogen, 12571-063, Waltham, MA, USA) supplemented with 10% *v/v* FBS and 1% *v/v* P/S.

#### 4.1.4. Gene Expression Analysis Primer Sequences

All gene expression primer sequences originate from the Harvard-Mass General Primer Bank (MGH-PGA, https://pga.mgh.harvard.edu/primerbank/ (accessed on 25 February 2022)).

### 4.2. Fabrication of Nanofibrous, Macroporous Tissue Engineering Scaffolds

Nanofibrous, macroporous tissue engineering scaffolds of various pore sizes were fabricated from poly (L-lactic acid), PLLA, as previously described [28,65]. In short, D-fructose sugar was melted and emulsified in hot mineral oil containing Span80 surfactant, with a magnetic stir bar. The fructose-mineral oil emulsion was cooled rapidly in an ice bath and solid sugar spheres were washed with hexane to remove residual oil and surfactant. The heterogeneous mixture of sugar spheres was purified by size using molecular sieves (Newark Wire Cloth Co, Newark, NJ, USA) to select for sugar spheres in the desired size ranges (Small: 60–125 μm, Medium: 125–250 μm, and Large: 250–425 μm). Meanwhile, PLLA was dissolved in THF at 10% *w/v* at 63 °C.

Sugar spheres of the desired size range were loaded into a Teflon mold, in hexane, and annealed at 37°C to cause partial adhesion to neighboring spheres (Small: 7 min, Medium: 9 min, Large: 12 min). Hexane was removed and the sugar template was dried under vacuum at room temperature. The PLLA/THF solution was cast and immediately stored at −80 °C to induce thermally induced phase separation (TIPS) for 48 h. After this time, the Teflon vials were submerged in hexane for 24 h, then in distilled water for 24 h to leach the sugar spheres. Resulting scaffolds were cut with a biopsy punch (5 mm diameter × 1.5 mm height for all cell and animal experiments reported herein) and were stored at −80 °C.

Prior to use, scaffolds were lyophilized to remove water and residual solvent for at least 24 h and were sterilized with ethylene oxide gas (Anpro, Haw River, NC, USA). Secondary sterilization was performed immediately prior to cell seeding by washing scaffolds in 70% ethanol (30 min), followed by washing three times with sterile phosphate buffered saline (PBS) and once with cell culture media containing FBS and antibiotics.

### 4.3. Scanning Electron Microscopy (SEM)

Scaffold samples were fixed to sample holders using two-sided electron microscopy-grade carbon tape, and were gold coated for 120 s (DeskII, Denton Vacuum, Morestown, NJ, USA). Observation was carried out at 5 kV with a working distance of 10–15 mm (JEOL JSM-7800 FLV). Note: Figure 7B shows an artificially colorized SEM image to highlight the location and distribution of MSCs cultured on the MMH tool surface.

### 4.4. Cell Isolation

All animal procedures were carried out under an approved protocol by the University of Michigan Institutional Animal Care and Use Committee (PRO00009613). We previously characterized their character as MSCs by gene expression and protein analysis [28].

#### 4.4.1. Bone Marrow Mesenchymal Stem Cells

Bone marrow mesenchymal stem cells (BMSCs) were isolated from wild type mice with no transgenic mutations on a C57BL/6J background, age 3–4 weeks old, based on a published protocol [28,80]. Mice were euthanized by CO_2_ asphyxiation and bilateral pneumothorax puncture. Tibiae and femur bones were dissected and harvested, cleaned from muscle and soft tissue, and kept in sterile PBS on ice. The ends of tibiae and femur bones were cut using a scalpel or surgical scissors, and the bone marrow was removed by centrifugation (5000 rpm, 5 min) using a 0.5 mL microcentrifuge tube and 1.5 mL microcentrifuge tube as described by Amend et al. [81]. Bone marrow aspirate was resuspended in cold, sterile PBS, filtered using a 40-μm cell strainer and washed with PBS three times. Primary BMSCs were cultured in DMEM containing 10% FBS and 1% P/S. All experiments used cells before passage 3.

#### 4.4.2. Suture Mesenchymal Stem Cells

Suture mesenchymal stem cells (SMSCs) were isolated from wild type mice with no transgenic mutations on a C57BL/6J background, age 3–5 days old, adapted from a published protocol [82]. The sagittal suture was dissected and removed from underlying tissues, then digested with 0.2% collagenase in sterile PBS at 37 °C for 1 h. Dissociated cells were filtered through a 40-μm cell strainer and washed with PBS. Primary SMSCs were cultured in αMEM containing 10% FBS and 1% P/S and were used before passage 4.

### 4.5. Cell Culture on Scaffolds

#### 4.5.1. Cell Seeding

Scaffold constructs (5 mm diameter × 1.5 mm height) were sterilized in 70% ethanol as described above. Primary cells cultured to confluence were trypsinized (0.05% trypsin, 37 °C, 3–5 min), washed, and concentrated to 16.6 million per mL in culture media. Additionally, 2.0 × 10^5^ cells were seeded to each scaffold in aliquots of 6 uL per side, in a non-treated polystyrene 24-well plate (CellPro, ASI No. TPN1024-NT, Orange, CA, USA). Cell–scaffold constructs were left for one hour at 37 °C to facilitate cell adhesion, then sufficient media was added to cover the constructs.

#### 4.5.2. Cell Culture

All cell–scaffold constructs were cultured in growth media; the composition was dependent on cell type, according to our previous work and established protocols [28]. BMSCs were cultured in DMEM containing 10% FBS and 1% P/S. SMSCs were cultured in αMEM containing 10% FBS and 1% P/S. Media was changed every two days.

#### 4.5.3. Administration of Cytochalasin D

Cytochalasin D (Cayman Chemical, No. 11330, Ann Arbor, MI, USA) was reconstituted in DMSO at a concentration of 100 mg/mL and stored in aliquots at −20 °C. Cytochalasin D was added to growth media (Gibco DMEM, 10% FBS, 1% P/S).

#### 4.5.4. Administration of Verteporfin

Verteporfin (Fisher, No. 50-202-9387, Waltham, MA, USA) was constituted in DMSO at 2 mg/mL and stored in aliquots at −20 °C. Verteporfin was added to growth media (Gibco DMEM, 10% FBS, 1% P/S), prepared fresh daily (VP, 5 μM).

#### 4.5.5. Administration of Lysophosphatidic Acid

Oleoyl-L-α-lysophosphatidic acid (Fisher, No. J66836LB0, Waltham, MA, USA) was constituted in sterile water (5 mg/mL) and stored in aliquots at −20 °C. Lysophosphatidic acid was added to growth media (Gibco DMEM, 10% FBS, 1% P/S), prepared fresh daily (LPA, 25 μM).

### 4.6. Subcutaneous Implantation in Mice

Cell–scaffold constructs, described in Section 4.5.1 (2 × 10^5^ cells/scaffold, 5 mm diameter scaffold × 1.5 mm height), were cultured for 24 h in vitro prior to their subcutaneous implantation into wild type mice, aged 8–10 weeks old. All animal procedures were carried out under an approved protocol by the University of Michigan Institutional Animal Care and Use Committee (PRO00009613). Mice were anesthetized with isoflurane via inhalation; a midsagittal incision was made on the dorsa, approximately 2 inches long. Four subcutaneous pockets were created in each quadrant of the dorsa by blunt dissection with surgical scissors, and one scaffold was implanted in each pocket. Incisions were closed with surgical stables and postoperative carprofen was given for pain management. Mice were closely monitored for postsurgical complications. After four weeks, mice were euthanized, and samples were explanted and fixed in 4% paraformaldehyde.

### 4.7. Histologic Preparation

Tissue samples from in vivo and in vitro experiments were fixed in 4% paraformaldehyde and dehydrated in 70% ethanol, then embedded in paraffin wax. Serial sections were cut at 5 μm thickness. Standard protocols were followed for hematoxylin and eosin (H&E) staining. Immunofluorescence staining is described below.

### 4.8. Quantitative Real Time Polymerase Chain Reaction (Gene Expression Assay)

cDNA was synthesized from RNA using SuperScript II cDNA Synthesis Kit (Invitrogen, Waltham, MA, USA) according to the manufacturer’s protocol. Quantitative real time polymerase chain reaction (rt-qPCR) was performed using Power SYBR Green PCR Master Mix (Applied Biosystems, Waltham, MA, USA), with Applied Biosystems ViiA7 platform. Gene expression was normalized to endogenous GAPDH expression. Primers are shown in Appendix A. Additional was performed with Morpheus (Broad Institute, https://software.broadinstitute.org/morpheus/, accessed on 15 December 2021) to generate heat maps and perform hierarchical clustering analysis.

### 4.9. Pore-Mimetic Imaging Platform for Three-Dimensional Visualization

Macropore-mimetic hemispheres (MMH) were designed in Tinker CAD (https://www.tinkercad.com/, accessed on 15 December 2021) with pore sizes that matched the average pore sizes determined by SEM image analysis (Figure 1B) and were exported as an STL file (Appendix A–C). MMH tools were 3D printed at the Duderstadt Center Fabrication Studio at the University of Michigan using a Stratasys J750 Polyjet 3D Printer (Rehovot, Israel) with 27 μm laser resolution and rigid opaque photopolymer.

Commercially available polydimethylsiloxane (PDMS), Sylgard 527 gel and Sylgard 184 elastomer (Dow Corning, Midland, MI, USA) were blended to match the mechanical properties of our PLLA scaffold materials. The ratio of Sylgard 184 to Sylgard 527 was 1:8, by weight [83]. The mixture was degassed in a desiccator and cast at room temperature, then cured for 24 h at 65 °C.

Prior to cell seeding, constructs were sterilized with ethylene oxide and 70% ethanol solution. Normal cell culture protocols were followed otherwise.

### 4.10. Immunohistochemistry

#### 4.10.1. Immunohistochemistry for Fluorescence Microscopy

Histologic sections prepared from paraffin were treated with xylene and an ethanol gradient to de-paraffinize. Antigen retrieval was performed by incubation in 10 mM citrate buffer (pH 6.0). Sections were sequentially incubated in 5% BSA for 60 min and with primary antibodies (tYAP, pYAP: 1:100; CD146: 1:200) overnight at 4 °C. Alexa Fluor 598 anti-rabbit IgG (1:200, Invitrogen, Waltham, MA, USA) was used as a secondary antibody. Slides were treated with 1:200 AlexaFluor 488 Phalloidin (Invitrogen, Waltham, MA, USA) and ProLong Gold antifade with DAPI (Invitrogen, Waltham, MA, USA). The same protocol was followed for immunohistochemistry in MMH constructs, without deparaffinization and antigen retrieval. Sections were observed by confocal laser microscopy.

#### 4.10.2. Western Blot

Whole cell lysates were separated on 4–20% Tris-Glycine polyacrylamide gel, then transferred to PVDF membranes which were incubated with 5% bovine milk and one hour, then primary antibodies overnight at 4 °C, listed above (tYAP, pYAP: 1:1000). Blots were incubated with peroxidase-coupled anti-rabbit IgG (1:2000, Cell Signaling Technologies, No. 7074) for one hour and protein expression was detected by Super Signal West Pico Chemiluminescent Substrate (Thermo Scientific, No. 34577, Waltham, MA, USA). Membranes were re-stained with anti-GAPDH (1:2000) to control for equal loading.

### 4.11. Confocal Laser Microscopy

Confocal laser microscopy was performed using a Nikon Eclipse C1 at the University of Michigan School of Dentistry Microscopy Core Facility.

### 4.12. Image Analysis

All image analysis was performed using Fiji (Image J, V 1.0.0-rc-69/1.52p). Statistical analysis and data visualization was performed in Prism.

#### 4.12.1. Analysis of Macropores from SEM Images

Images of scaffold morphology were acquired by scanning electron microscopy (SEM). The scale of images was calibrated to the magnification of the microscope to convert pixel distances to microns. Distances were measured across the longest cord spanning a macropore, for each pore in the field of view. Measurements were made across a minimum of n = 15 images per group.

#### 4.12.2. Spatial Point Pattern Analysis

Spatial point pattern analysis was performed in R (version 4.1.2), by a method described in the literature [54] and the spatstat package [53], taking examples of spatial point pattern analysis for ecological application. Histologic images from sections prepared from subcutaneous explants were stained with hematoxylin and eosin staining and imaged. Images were imported to Fiji (Image J, see above), converted to a binary image and a threshold was applied to identify nuclei based on H&E stain; each nucleus was converted to a point and the (x, y) coordinate locations of each event were exported and converted to a pattern of points in R (>plot(mypattern)), for statistical analysis. A minimum of n = 15 images were analyzed for each group from n = 4 explants per group. A data frame boundary was implemented to control for space occupied by the polymer matrix (appears white in histology) and unavailable for cells to occupy.

The F-function (>plot (Fest(mypattern)) measures the distribution of all distances from an arbitrary point in the plane to the nearest observed event:$$F\left(r\right)=1-e^{-\lambda \pi r^{2}}.$$

Ripley’s K function (>plot (Kest(mypattern)); <plot (envelope (mypattern, Kest))) was computed to calculate the probability of points to appear clustered, independently, or regularly spaced:$$K\left(s\right)=\lambda^{-1}E\left[N_{0}\left(s\right)\right].$$

#### 4.12.3. Nuclear Morphology in Scaffold Sections

Immunofluorescence images from confocal laser microscopy were imported to Fiji and converted to an RGB Stack. The “blue” layer corresponds to DAPI+ staining of cell nuclei. The “blue” layer was converted to binary, and the height and width of each nuclei was measured. Nuclear aspect ratio (*NAR*) was calculated as:$$NAR=\frac{d_{short\;axis}}{d_{long\;axis}}.$$

Nuclear cross-sectional area was calculated from binary images of DAPI staining by measuring the DAPI+ area by threshold, and the number of nuclei by cell counting, calculated:$$Cross-Sectional\;Area=\frac{DAPI+signal}{\#\;cells}.$$

#### 4.12.4. Quantification of Cell Clustering in Scaffold Sections

Immunofluorescence images from confocal laser microscopy were imported to Fiji and converted to an RGB Stack. The “blue” layer corresponds to DAPI+ staining of cell nuclei. The “blue” layer was converted to binary, and the number of cells per discrete cell cluster was counted and recorded. Then the distance between each nuclei and its nearest neighbor was measured.

#### 4.12.5. Quantification of YAP-Nuclear Colocalization

Immunofluorescence images from confocal laser microscopy were imported to Fiji and converted to an RGB Stack. The “red” layer represents the tYAP+ signal; the “blue” layer represents the DAPI+ signal. The boundary of each object in the blue channel (each nucleus) was traced using the freehand drawing tool, and the red+ signal in the DAPI+ boundary region as measured, and its intensity was recorded for each nucleus. Then, the total red+ signal per cell was measured. The tYAP nuclear colocalization ratio was calculated as:$$Coloc=\frac{A_{red+within\;blue+boundary}}{A_{total\;red+}}.$$

#### 4.12.6. Quantification of YAP Phosphorylation

Immunofluorescence images from confocal laser microscopy were imported to Fiji and converted to an RGB Stack. The “red” layer represents the pYAP+ signal; the “blue” layer represents the DAPI+ signal. A threshold was applied to the “red” layer and the area of red+ signal was recorded; the number of nuclei in the “blue” layer (DAPI+ points) was assessed by cell counting. The normalized intensity of pYAP+ signal was calculated by:$$I_{pYAP+}=\frac{A_{red+}}{\#\;nucle\;in\;field}.$$

### 4.13. Statistical Analysis

All data are reported as mean ± standard deviation and represent a minimum sample size of n ≤ 4. Statistical analysis was carried out in GraphPad Prism v9. Student’s *t*-test was used to determine the statistical significance of observed values between experimental groups where *p* < 0.05 was considered significant. Statistical analyses were carried out under the guidance of the University of Michigan Consulting for Statistics, Computational and Analytical Research Center. In all graphics, significance is noted as: * *p* < 0.05, ** *p* < 0.01, *** *p* < 0.001, **** *p* < 0.0001.

## 5. Conclusions

Mesenchymal stem cells are sensitive to their mechanical environment. Like growth factor treatment, the mechanical microenvironment may be similarly exploited as a fate-guiding cue in vitro and in vivo. We demonstrate that the principle curvature of spherical scaffold macropores is a critical biomaterial design criteria; sufficiently steep principle curvature in sufficiently small diameter pores <125 μm inhibits YAP/TAZ activation and favors cytosolic YAP phosphorylation (inactivation) responsible for the maintenance of stemness, while sufficiently shallow principle curvature facilitates YAP/TAZ activation and nuclear translocation as early as 48 h after seeding and is, in part, responsible for curvature-mediated osteogenic differentiation trajectories. Highly controlled scaffolding technologies for biomaterials-mediated regeneration can be designed to modulate MSC differentiation fate through an intimate understanding of the cell–biomaterial interface and may be useful for increasing the predictability of tissue-specific regenerative interventions in future developments of scaffolding technology.

## Data Availability

All data associated with this work are in the figures or Appendix A.

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
