# Peer review of "Scaffold Pore Curvature Influences ΜSC Fate through Differential Cellular Organization and YAP/TAZ Activity"

_ijms, 2022, doi:10.3390/ijms23094499_

Round 1
Reviewer 1 Report
Dear authors, the topic of great interest in this field and cell functionalized scaffolds represent a suitable tool for the development of regenerative therapies. Also the patwhay YAP / TAZ is getting attention. To obtain the results many techniques have been used and it is certainly a plus. However, to better explain your experimental setup you could produce a schematic figure that represents it. Also why did you choose 4 weeks as a time point for the explant? Moreover, the in vitro section is extensively treated, while less attention is given to the in vivo section. You should emphasize this part. Also have you tested the cell-free scaaffold in vivo to see if it is able to recall cells from the host and if there can be any type of cell contamination compared to those seeded?Author Response
March 16, 2022
Re: ijms-1635854, revised manuscript
Dear Mr. Chen,
We are very grateful for your kind invitation to contribute to your special issue, “Advanced Bioscaffolds as Drivers of Modern Medicine.” Based on reviewers’ comments, we have revised our manuscript entitled, “Scaffold Pore Curvature Influences MSC Fate through Differential Cellular Organization and YAP/TAZ Activity,” for consideration in International Journal of Molecular Sciences. We confirm that this work is being exclusively submitted to IJMS and has not been published previously nor will be published elsewhere in the same form. All contributing authors have reviewed and approved of the manuscript and have no competing interests.
Herein please find a revised manuscript and point-by-point description of the revision included in this submission. The manuscript was revised using the “Track Changes” function in MS Word, as requested. We are grateful to the reviewers for their kind remarks and helpful critique of our manuscript. These comments have helped us to communicate our exciting findings more articulately to the broader community and helped to strengthen the quality of this work.
Thank you for your efforts in evaluating our revised manuscript for publication in IJMS.
Sincerely,
Prof. Yuji Mishina
Department of Biologic and Materials Sciences & Prosthodontics
University of Michigan, School of Dentistry
Response to Reviewers
ijms-1635854
Reviewer 1:
Dear authors, the topic of great interest in this field and cell functionalized scaffolds represent a suitable tool for the development of regenerative therapies. Also the pathway YAP / TAZ is getting attention. To obtain the results many techniques have been used and it is certainly a plus.
We thank the author for their kind review of the manuscript and enthusiasm for the topic.
However, to better explain your experimental setup you could produce a schematic figure that represents it.
Thank you for this suggestion. We have also added a brief schematic to Figure 2 to describe the subcutaneous implantation study. We believe that in combination with Scheme 3 and Figure 11 schematics, this new graphic provides a more comprehensive succinct review of the major experiments and outcomes assessed.
Also why did you choose 4 weeks as a time point for the explant?
4 weeks’ time was chosen as an endpoint for our subcutaneous implantation because we previously observed significant differences in tissue gene expression and phenotype (vascularization, extracellular matrix composition) at this time in our previous studies (Swanson et al, Biomaterials 2021). Given differences in tissue fate specification, we were eager to investigate organizational differences at this time point. We added a statement to Section 2.2 to clarify this rationale.
Moreover, the in vitro section is extensively treated, while less attention is given to the in vivo section. You should emphasize this part. Also have you tested the cell-free scaffold in vivo to see if it is able to recall cells from the host and if there can be any type of cell contamination compared to those seeded?
Previously we reported that scaffold pore size modulates cell and tissue phenotype and genotype in vivo, summarized in Scheme 1. These findings were confirmed in vitro, leading to the hypothesis that scaffold macropore size is a key driver of cell fate, irrespective of the in vivo environment. Our summary of these findings provide reference to Swanson and Gupte et al, Acta Biomaterialia 2018 and Swanson et al, Biomaterials 2021. The in vivo data presented in the present manuscript examines the hypothesis that scaffold pore size affects cell organization within the cell-scaffold construct, which we summarized in Figure 2. These in vivo studies of cell organization served as a proof of concept that scaffold pore size affects cell fate via differential organization, which we probed using in vitro methods in subsequent experiments. We have added a statement to Section 2.2 to emphasize our thought process and clarify how the in vivo study lead into the in vitro mechanistic investigation which comprises the majority of the paper.
The reviewer provides an excellent suggestion to examine the fate of endogenous cells which migrate into the scaffold to determine the degree to which scaffold pore size modulates their fate. These studies are currently underway in our group, and we look forward to sharing these findings with the community.
Reviewer 2:
The commends from Reviewer 2 are associated with a different manuscript (Trevisani et al) and therefore not actionable. We communicated this with Mr. Chen and revised the paper according to Reviewer 1’s report.

Reviewer 2 Report
Swanson’s at al. manuscript represents an in-depth study of the effect of scaffold pore curvature on the MSC differential cellular organization. The research is fundamental and very strong, and undoubtedly merits to be published in IJMS. The manuscript is clearly written, discussed and well-illustrated. I recommend it for publication after minor revision.
- I have not found any information on the dimensions of porous scaffolds used in the experiment with cells. Please provide while presenting results on cell experiments.
- Scheme 1. Please provide a Ref. in parentheses as a number according to the list of reference.
- Lines 146-148. Please place Figure 1 before Scheme 2 as they mentioned in the text.
- Figure 1 and following. Please replace “um” units with “µm” ones.
- Please unify the volume units in the Experimental part (e.g. line 463 1.6 dl/g while line 554 M/mL).
Author Response
April 8, 2022
Re: ijms-1635854, revised manuscript
Dear Mr. Chen,
We are very grateful for your kind invitation to contribute to your special issue, “Advanced Bioscaffolds as Drivers of Modern Medicine.” Based on reviewers’ comments, we have revised our manuscript entitled, “Scaffold Pore Curvature Influences MSC Fate through Differential Cellular Organization and YAP/TAZ Activity,” for consideration in International Journal of Molecular Sciences. We confirm that this work is being exclusively submitted to IJMS andhas not been published previously nor will be published elsewhere in the same form. All contributing authors have reviewed and approved of the manuscript and have no competing interests.
Herein please find a revised manuscript and point-by-point description of the revision included in this submission. The manuscript was revised using the “Track Changes” function in MS Word, as requested. This revision incorporated the feedback from Reviewer 3. We are grateful to the reviewers for their kind remarks and helpful critique of our manuscript. These comments have helped us to communicate our exciting findings more articulately to the broader community and helped to strengthen the quality of this work.
Thank you for your efforts in evaluating our revised manuscript for publication in IJMS.
Sincerely,
Prof. Yuji Mishina
Department of Biologic and Materials Sciences & Prosthodontics
University of Michigan, School of Dentistry
Response to Reviewers
ijms-1635854
Reviewer 1:
Dear authors, the topic of great interest in this field and cell functionalized scaffolds represent a suitable tool for the development of regenerative therapies. Also the pathway YAP / TAZ is getting attention. To obtain the results many techniques have been used and it is certainly a plus.
We thank the author for their kind review of the manuscript and enthusiasm for the topic.
However, to better explain your experimental setup you could produce a schematic figure that represents it.
Thank you for this suggestion. We have also added a brief schematic to Figure 2 to describe the subcutaneous implantation study. We believe that in combination with Scheme 3 and Figure 11 schematics, this new graphic provides a more comprehensive succinct review of the major experiments and outcomes assessed.
Also why did you choose 4 weeks as a time point for the explant?
4 weeks’ time was chosen as an endpoint for our subcutaneous implantation because we previously observed significant differences in tissue gene expression and phenotype (vascularization, extracellular matrix composition) at this time in our previous studies (Swanson et al, Biomaterials 2021). Given differences in tissue fate specification, we were eager to investigate organizational differences at this time point. We added a statement to Section 2.2 to clarify this rationale.
Moreover, the in vitro section is extensively treated, while less attention is given to the in vivo section. You should emphasize this part. Also have you tested the cell-free scaffold in vivo to see if it is able to recall cells from the host and if there can be any type of cell contamination compared to those seeded?
Previously we reported that scaffold pore size modulates cell and tissue phenotype and genotype in vivo, summarized in Scheme 1. These findings were confirmed in vitro, leading to the hypothesis that scaffold macropore size is a key driver of cell fate, irrespective of the in vivo environment. Our summary of these findings provide reference to Swanson and Gupte et al, Acta Biomaterialia 2018 and Swanson et al, Biomaterials 2021. The in vivo data presented in the present manuscript examines the hypothesis that scaffold pore size affects cell organization within the cell-scaffold construct, which we summarized in Figure 2. These in vivo studies of cell organization served as a proof of concept that scaffold pore size affects cell fate via differential organization, which we probed using in vitro methods in subsequent experiments. We have added a statement to Section 2.2 to emphasize our thought process and clarify how the in vivo study lead into the in vitro mechanistic investigation which comprises the majority of the paper.
The reviewer provides an excellent suggestion to examine the fate of endogenous cells which migrate into the scaffold to determine the degree to which scaffold pore size modulates their fate. These studies are currently underway in our group, and we look forward to sharing these findings with the community.
Reviewer 2:
The commends from Reviewer 2 are associated with a different manuscript (Trevisani et al) and therefore not actionable. We communicated this with Mr. Chen and revised the paper according to Reviewer 1’s report.
Reviewer 3:
Swanson’s at al. manuscript represents an in-depth study of the effect of scaffold pore curvature
on the MSC differential cellular organization. The research is fundamental and very strong, and
undoubtedly merits to be published in IJMS. The manuscript is clearly written, discussed and
well-illustrated. I recommend it for publication after minor revision.
We thank Reviewer 3 for his kind support of our manuscript and appreciate their suggestions, which we address below.
- I have not found any information on the dimensions of porous scaffolds used in the
experiment with cells. Please provide while presenting results on cell experiments.
We have added the dimensions of the scaffolds in both the methods (in vitro and in vivo work), and results sections. The scaffolds are 5 mm diameter x 1.5 mm height.
- Scheme 1. Please provide a Ref. in parentheses as a number according to the list of
reference.
We have added the numbered reference as suggested.
- Lines 146-148. Please place Figure 1 before Scheme 2 as they mentioned in the text.
We have rearranged the figures in the text as suggested.
- Figure 1 and following. Please replace “um” units with “μm” ones.
We have revised the figures as suggested and updated in the manuscript.
- Please unify the volume units in the Experimental part (e.g. line 463 1.6 dl/g while line 554
M/mL).
We have revised M/mL to read “million per mL” as a measure of cell number per volume of culture media in our scaffold seeding protocol. “dl/g” has been changed to “dL/g” and is a measure of inherent viscosity reported by the manufacturer.